# Study of Behavioural Traits in Can de Palleiro (Galician Shepherd Dog)

**DOI:** 10.3390/ani11113198

**Published:** 2021-11-09

**Authors:** Susana Muñiz de Miguel, Francisco Javier Diéguez, Joao Pedro da Silva-Monteiro, Beatriz Parra Ferreiro-Mazón, Ángela González-Martínez

**Affiliations:** 1Servicio de Etología y Medicina de Comportamiento Animal, Hospital Veterinario Universitario Rof Codina, Facultad de Veterinaria, Universidad de Santiago de Compostela, 27002 Lugo, Spain; susana.muniz.demiguel@usc.es (S.M.d.M.); beatriz.parra@rai.usc.es (B.P.F.-M.); 2Departamento de Anatomía, Producción Animal y Ciencias Clínicas Veterinarias, Facultad de Veterinaria, Universidad de Santiago de Compostela, 27002 Lugo, Spain; franciscojavier.dieguez@usc.es; 3Bom Jesus Veterinary Hospital, 4715-380 Braga, Portugal; joaopedromonteiro@outlook.com

**Keywords:** canine, autochthonous breeds, comportment assessment, C-BARQ, SAB, ordinal regression

## Abstract

**Simple Summary:**

The Can de Palleiro or Galician shepherd is a canine breed that was in danger of extinction but is currently growing rapidly in popularity. In this study, different behavioural traits of the breed were evaluated in order to assess breeds, select the best breeding animals and identify behaviour problems. This is the first study carried out in the Can de Palleiro breed using different scientifically validated tests. Questionnaires filled by the owners (C_BARQ) were collected, and a behavioural test (SAB) was conducted to evaluate the response of the dogs to a specific stimulus at a certain time and in a certain environment. In addition, the results from the Can de Palleiro breed were compared with those obtained from the general canine population of Galicia. Thereby, the Can de Palleiro breed showed less owner-directed aggression, dog-directed fear, excitability, non-social fear and separation-related problems and better trainability.

**Abstract:**

The Can de Palleiro (CP) is an autochthonous canine breed from Galicia (NW Spain). Interestingly, no previous research has been published about the behaviour of this breed. Thus, the aim of the present study was to obtain a deeper understanding of CP behavioural and temperamental traits and detect any potentially problematic behaviour by using the Canine Behavioural Assessment and Research Questionnaire (C-BARQ) and the Socially Acceptable Behaviour (SAB) test. Behavioural information was obtained from 377 dogs—177 CPs and 200 general population (GP) dogs—using the C-BARQ. Additionally, 32 dogs were enrolled to perform the SAB test (19 CPs and 13 GP dogs) in order to directly evaluate their temperament. Our results indicated that CP dogs had a lower tendency to show aggressiveness towards their owners (0.18 times lower, *p* = 0.033) and less fear of other dogs (by 0.43 times, *p* = 0.001), as well as higher trainability levels (2.56 times higher, *p* < 0.001) when compared to GP dogs. CP dogs also had increased odds of showing chasing behaviour (3.81 times higher, *p* < 0.001). Conversely, CPs had reduced odds of non-social fear, separation-related problems and excitability (by 0.42, 0.35 and 0.48 times, respectively; *p* < 0.001, *p* < 0.001 and *p* = 0.002). The current research represents a starting point for the study of the behaviour of CPs, which appear to be a working breed, with guarding and, especially, herding characteristics.

## 1. Introduction

Can de Palleiro (CP) is an autochthonous canine breed from Galicia (NW Spain), also known as the Galician Shepherd. The CP is of Indo-European descent, which is evident from its rustic characteristics, with its ancestors likely being native to Galicia, having been imported by Galicians during the Palaeolithic era while expanding from the British Isles and European continent. The breed shares some of its origin story with German Shepherds, Belgian Shepherds, Dutch Shepherds and the Portuguese Castro Laboreiro.

The CP is a rustic and lupine-type dog with a straight profile which is eumetric and mesodolichomorphic; it is of medium size, with a height of about 60–62 cm at the withers, harmonic proportions, a strong constitution, and fairly wide (‘thick-set’) bones. Females are somewhat shorter and lighter in appearance. In general, CPs are considered to have a strong and reticent character with strangers but usually show great loyalty to their owners with whom they are docile and calm [1,2,3].

The breed was traditionally employed in the rural areas of Galicia as herding dogs and as guard dogs at farmers’ houses where they would sleep in the hayloft, or palleiro. However, since the middle of the 20th century, a significant reduction in the rural population of CPs and the introduction of foreign breeds has put CPs in danger of becoming extinct [1,4].

At the very end of the 1990s, the regional government in Galicia decided to study the possibility of reviving the breed and started by searching for surviving specimens. Thus, an official breed standard was described, which allowed a recovery project aimed at increasing the number of CP individuals. In addition, the CP club was created, which has been guardian of the breed since 2002 by overseeing all the official offspring and by organising adoptions to help revive the Galician Shepherd [3,4]. Nonetheless, CP remains a particularly vulnerable dog, with only 1775 specimens currently registered in the herd book (unpublished data). Moreover, in addition to its characteristic shepherd function, its employment in activities related to police or rescue services units is now being contemplated.

Although the breed standard refers to the temperament and behaviour of CPs, to date, no studies have used validated tools to evaluate these traits [3]. Gathering scientific knowledge on the behavioural characteristics of dog breeds has both scientific and applied purposes [5]. Behaviour is driven by a complex interaction between endocrine and neuroendocrine factors [6]. In turn, these factors are influenced by both genetics and the environment. In fact, it was shown that several behavioural problems such as fear or aggression have a genetic component [7,8,9]. Thus, it is important to carefully choose breeding individuals to try to avoid undesirable behaviours in future generations. In this sense, understanding the behaviour of a breed is likely the first step for promoting more responsible breeding.

Several standardised tests to evaluate dog behaviour and temperament have already been developed and validated. In general, a sample of dogs is subjected to the same or similar stimuli as part of these tests, while human observers attempt to measure several behaviours [10,11,12,13,14]. Although the Socially Acceptable Behaviour (SAB) test was originally developed as a test of aggression toward unfamiliar people, this tool has also been validated to assess aggressive and fearful behaviours in dogs by analysing their response to 16 standardised subtests [10,14,15,16,17].

Indeed, it has also been shown that the SAB test can help to reduce unwanted behaviours in dog populations when used to direct breeding policies [18]. Nonetheless, although this test is comprehensive, it is often difficult to conduct, and the possibility that the experimental setting itself could result in the emergence of novel behaviours cannot be excluded [19].

Other methods focus on the assessment of day-to-day behaviour using a questionnaire for dog owners. They are often used as a means of validating behavioural tests and typically provide detailed information about a given dog’s tendency to display different behaviours because owners can observe their animals in a variety of situations over an extended period. In this sense, a widely used questionnaire is the canine behavioural assessment and research questionnaire (C-BARQ). The C-BARQ contains questions regarding aggression, fear and anxiety, trainability, excitability, separation-related behaviour, attachment, attention seeking, and chasing [20,21,22]. It has been used with different purposes, including the comparison of the behavioural profile of different breeds [19,22], to study civilian working dogs [23], military working dogs [24], drug detection dogs [25] and guide and service dogs [26].

Based on the above, the aim of this present article was to use the C-BARQ and SAB tests to assess the behavioural traits of CP and to compare the results with those obtained for the general canine population.

## 2. Material and Methods

### 2.1. Surveyed Animals 

This study was carried out at the Rof Codina Veterinary Teaching Hospital (VTHRC) that is part of the Veterinary Faculty of Lugo and is the referral centre for veterinary clinics in Northwest Spain. We included 377 dogs, 177 (46.9%) of which were CPs. The CP dogs were recruited through the Can de Palleiro Club with which the VTHRC has a collaboration agreement. The remaining 200 general population (GP) dogs (53.1%) were recruited through social media networks such as Facebook, Instagram, and LinkedIn and by sharing an e-mail chain, starting with our personal contacts. The GP dogs were dogs of different breeds (except CP) and were used as a representative sample of the general Galician canine population (Table 1).

### 2.2. Behaviour Assessment and Data Collection

Behavioural information about the 377 dogs was obtained using the C-BARQ, a validated questionnaire developed by Hsu and Serpell (2003) [18]. Additionally, a randomly selected sample of 32 dogs from the overall population (19 CPs and 13 GPs, including 1 miniature Pinscher, 1 Dachshund, 1 Poodle, 1 German Shepherd, 1 Podenco and 8 mixed-breed dogs of different morphotypes) completed the SAB test [10] to directly evaluate the temperament of these animals. Descriptive statistics summarising the characteristics of the 377 dogs, as well as the results from the 32 dogs that undertook the SAB test, are presented in Table 2.

The C-BARQ was filled in by the owners via an online platform. The owners were able to contact one of the veterinarians responsible for the ethology service at the HCVRC to discuss and solve any possible doubt or misunderstanding they had regarding the C-BARQ. This questionnaire comprises 100 questions which describe different ways in which dogs typically respond to common events, situations, and stimuli in their environment. The responses are grouped into 14 behavioural traits as follows: stranger-directed aggression, owner-directed aggression, dog-directed aggression, dog-directed fear, familiar dog aggression, trainability, chasing, stranger-directed fear, non-social fear, separation-related problems, touch sensitivity, attachment/attention seeking, excitability and energy [20,21,22] (Appendix A
Table A1). All the traits are expressed on a 0 to 4 scale, in which 0 indicates no sign of the behaviour in question, and 4 indicates the presence of a severe form of the behaviour.

The SAB test was set up by following the procedure described by Planta and De Meester (2007) [10] and was performed at the HCVRC by one of the authors (S.M.M.). The test comprises 16 subtests which analyse posture and behavioural responses to different stimuli which are displayed in a fixed order [10,15,16,17] (Appendix A
Table A2). Thus, posture and behavioural strategy scores (Appendix A
Table A3 and Table A4) were recorded according to De Meester et al. (2011) [15]. The neutral position was defined as the posture adopted in an active but relaxed mood according to the breed standard.

### 2.3. Statistical Analysis

All the statistical tests were performed using SPSS 15.0 software (SPSS Inc., Chicago, IL, USA). The effect of breed (GP vs. CP) on the C-BARQ scores for the different behavioural traits was examined by applying the Kruskal–Wallis test. The effect of breed on the responses observed through the SAB test was assessed using the Fisher exact test. In this test, to assess the agreement between observers in the positions and strategies shown by the dogs for each subtest, the kappa (κ) index was used.

Afterwards, ordinal regression models were fitted to assess the effect of breed (CP vs. GP) on the C-BARQ score for the different traits (given that data from all 377 dogs were available for this test). Likewise, the following explanatory variables were considered in the models fitted as control variables: gender, age, age at the time of acquisition, neutered vs. unneutered, dog activity pattern (if the dog displayed any exceptional activities or not, i.e., bike, walking, herding) and whether the owner had previously owned other dogs.

One model was tested for each of the following C-BARQ traits: stranger-directed aggression, owner-directed aggression, dog-directed aggression, dog-directed fear, familiar dog aggression, trainability, chasing, stranger-directed fear, non-social fear, separation-related problems, touch sensitivity, attachment/attention seeking, excitability and energy.

The scores obtained for each of the traits were divided into five categories [27]:-0: C-BARQ score = 0.-1: C-BARQ score >0 to 1.-2: C-BARQ score >1 to 2.-3: C-BARQ score >2 to 3.-4: C-BARQ score >3 to 4.

Therefore, the following odds were modelled [27]:-C-BARQ score 0, 1, 2, 3 vs. 4-C-BARQ score 0, 1, 2 vs. 3, 4-C-BARQ score 0, 1 vs. 2, 3, 4-C-BARQ score 0 vs. 1, 2, 3, 4

The ordinal regression model provided the odds ratios for higher levels of the C-BARQ score (relative to being in or below a given score). When a variable changed the effect of the remaining coefficients by 10% or more, it was considered a confounder, and we retained it in the model, regardless of its level of significance [27].

The parallel line test was used to assess the hypothesis of proportionality. Ordered logistic regression assumes that the coefficients that describe the relationship between the lowest versus the highest response variable categories are the same as those that describe the relationship between the next lowest category and all the higher categories [28].

## 3. Results

### 3.1. Canine Behavioural Assessment and Research Questionnaire Results

The mean and median C-BARQ scores (along with quartiles) obtained for the CP and their counterparts for the GP are provided in Table 3. Differences between groups (CP vs. GP dogs) were significant except for familiar dog aggression and attachment/attention seeking. Compared to GP dogs, CPs showed significantly higher scores for stranger-directed aggression, dog-directed aggression, trainability, chasing and energy, as well as significantly lower owner-directed aggression, dog-directed fear, stranger-directed fear, non-social fear, separation-related problems, excitability and touch sensitivity.

### 3.2. Socially Acceptable Behaviour Results

Two CP out of the 32 CP dogs for which the test was performed presented extreme fear during the test, and so the SAB was interrupted and cancelled for animal welfare reasons.

The kappa values to assess the agreement between the observers for the different postures and strategies displayed by the dogs for each subtest showed values >0.85 except for posture 7 in subtest 3 (κ = 0.74) and posture 3 in subtest 10 (κ = 0.78).

The following results from the SAB test were statistically significant. When the dogs were exposed to an unfamiliar sound, 69.2% of the GP dogs showed strategy 3, while only 26.3% of the CPs showed this behaviour. Additionally, 23.0% of the GP dogs showed postures compatible with extreme fear (score = 6) when they were approached by unfamiliar people who were staring at the dog, while none of the CPs (0%) showed this behaviour. Finally, when the owner tried to pet the dog with a doll, 100% of the GP dogs showed slight signs of stress (strategy 1), while only 70.6% of the CPs did.

### 3.3. Regression Models

According to the regression models, no relationship was found between the groups (CP vs. GP) for stranger-directed aggression, dog-directed aggression, familiar-dog aggression, stranger-directed fear, attachment/attention seeking, touch sensitivity or energy after correcting for the control variables.

Regarding owner-directed aggression, the ordinal model indicated that CPs had a C-BARQ score 0.18 times lower than the GP dogs (*p* = 0.033) for this factor. The factors of age and having previously owned other dogs were retained in the model because their exclusion modified the coefficients by more than 10%. The remaining control variables (i.e., gender, neutered vs. unneutered, dog activity pattern and age at the time of acquisition), being non-significant, were excluded from the model, since their elimination did not modify the remaining coefficients by more than 10%. The same procedure was followed in the subsequent models (Table 4).

Ordinal regression also indicated that CP dogs had a 3.0-fold lower score for dog-directed fear (*p* = 0.001) and a 2.56 times higher score for trainability (*p* < 0.001) than GP dogs. Trainability was also influenced by the neutering status, having previously owned other dogs, and the dog activity pattern (Table 4).

CP dogs had an increased odds ratio of a higher chasing score by a factor of 3.81 (*p* < 0.001). Conversely, CPs had reduced odds of non-social fear (0.42-fold, *p* < 0.001), separation-related problems (0.35-fold, *p* < 0.001) and excitability (0.48-fold, *p* = 0.002). Age had also a significant effect on non-social fear. The parallel line test, non-significant in the models fitted, confirmed the suitability of the ordinal regression models.

## 4. Discussion

The CP dog breed is rapidly growing in popularity as a breed of working and companion animals. However, scientific evidence of the behavioural characteristics of CPs is still very limited; therefore, to the best of our knowledge, this current study is the first investigative research on the behavioural profile of CPs.

To assess dog behaviour and tackle possible behavioural problems, it can be effective to obtain information about individual dogs from their owners, who usually best understand the typical behaviours of their dogs. Serpell and Hsu (2001) developed the C-BARQ instrument to measure behavioural traits in pet dogs [29]. The C-BARQ is a useful and validated resource for investigating dogs’ behaviour, and several studies have used it to examine and compare breed differences in behavioural traits, with previous findings [16,21,22,30].

In this research, C-BARQ was distributed between CP and GP owners. GP dogs included breeds of all the Federation Cynologique Internationale groups, as well as mixed-breed dogs of different morphotypes.

The Kruskal–Wallis test showed significant differences between the CP and the GP groups for stranger-directed aggression, dog-directed aggression, trainability, chasing, energy, owner-directed aggression, dog-directed fear, stranger-directed fear, non-social fear, separation-related problems, excitability and touch sensitivity. Nevertheless, when the breed was assessed in a multivariate model with other control variables that could also be risk factors for these behavioural problems (i.e., age, age at acquisition, sex, neutering, having previously owned other dogs and activity pattern [31,32,33,34,35], no differences were found for stranger-directed aggression, dog-directed aggression, familiar-dog aggression, stranger-directed fear, attachment/attention seeking, touch sensitivity or energy. However, one possible limitation of this study is that, given its limited sample size, multivariate models were not applied in the case of the SAB test.

According to the ordinal model, CP dogs had a significantly lower C-BARQ owner-directed aggression factor score than GP dogs. Similar results have also been found for other shepherd breeds [19,36] with which CPs share a common origin. Indeed, the breed standard [3] also reports that CPs are loyal and docile dogs with their owners. Like other shepherd and guarding breeds (i.e., Australian Shepherds or Rottweilers) [36], CPs tend to show less dog-directed fear and higher levels of trainability.

Interestingly, lower scores for excitability, non-social fear and separation-related problems were found for CP dogs, with these lower scores corresponding to a steady temperament as defined in the breed standard [3].

The C-BARQ chasing factor refers to the tendency of some dogs to display predatory chasing of cats, squirrels, birds and/or other small animals when given the opportunity. In this work, CPs obtained higher chasing scores than GP dogs. Nevertheless, most herding breeds strongly express predatory motor patterns such as stalking, while more advanced aspects of the canine hunting sequence (grabbing) are differentially developed among herding dogs. For instance, herding breeds such as the Australian cattle dog (which is used to working with typically stubborn cattle) strongly express grab-biting behaviours [37,38].

Even though only a few animals from the overall cohort performed the SAB test in this work, its results coincided with some of the C-BARQ findings. Thus, the GP group but not the CP group showed stronger avoidance behaviour towards unfamiliar sounds, which also matches with the differences in non-social fear found with the C-BARQ. Of note, CP dogs showed less fear when their owner tried to pet their dog with a doll than GP dogs; correspondingly, in the C-BARQ, CPs showed less aggression toward family members than GP dogs. However, the combination of an unfamiliar stimulus (the doll) with the owner, who is most likely associated with a positive emotional state in the dog, might have decreased the novelty effect of the doll, thereby increasing the dog’s capacity to maintain emotional homeostasis [39,40].

However, the SAB test showed significant differences between the groups for extreme fear when the dogs were approached by unfamiliar people staring at them, although no differences between the groups were found in the ordinal model for social fear using the C-BARQ. A possible explanation could be that this subtest was performed in the absence of the owner. The behaviour of both confident and fearful dogs can change when their owner is not present; both confident and fearful dogs can experience an increase in the posture score (lower postures), but either group may also react in the same way as if the owner were present. Therefore, it is impossible to predict a given dog’s behaviour in the absence or presence of its owner [15].

Control variables included in the ordinal models are considered risk factors for behaviour problems in dogs. Thereby, some research revealed that males have a higher risk for behavioural problems than females [20,31,32,41,42]. Nevertheless, in the present study, only excitability was influenced by sex. Controversial results were found in previous studies. In fact, Takeuchi et al. [43] and Bradshaw et al. [44] showed that males are more excitable than females, whereas other surveys found that females tend to be more excitable [45,46]. The role of gonadectomy on behavior is complex; indeed, it was used to treat some behaviour problems, such as urine marking, mounting, roaming and intrasexual aggression in male dogs [47,48,49]. However, similar to our results, some studies revealed neutering as a risk factor for fear, anxiety, aggression and even some cognitive alterations [35,50,51,52,53]. It has been suggested that these behaviour problems in neutered dogs are related to the continuous elevation of luteinizing hormone at supraphysiologic concentrations occurring in gonadectomized animals [54].

One limitation of this study is that the CP dogs in the SAB test were heavily female-skewed, and overall the dogs subjected to the SAB were skewed towards not neutered dogs; this could have influenced the results. More studies are necessary, with a higher and more homogeneous sample of dogs taking the SAB, to obtain more accurate results.

## 5. Conclusions

This current research represents a starting point for the study of CP behaviour. According to the differences in behavioural traits between CPs and the GP dogs we measured in this study, CPs seem to be a working breed with guarding and, especially, herding characteristics. As such, CPs exhibited lower scores for owner-directed aggression, dog-directed fear, excitability, non-social fear and separation-related anxiety and higher scores for chasing and trainability.

## Figures and Tables

**Table 1 animals-11-03198-t001:** Distribution of breeds in the sample deemed as the general Galician canine population.

Breed Group (FCI)	Breed	n	%	% Per Group
Group 1: Sheepdogs and Cattle dogs (except Swiss Cattle dogs)	Pastor Vasco *	7	3.5	12
	White Swiss Shepherd Dog	4	2	
	German Shepherd dog	5	2.5	
	Border Collie	6	3	
	Belgian Shepherd dog	2	1	
Group 2: Pinscher and Schnauzer–Molossoid and Swiss Mountain and Cattle dogs	Boxer	16	8	12
	French Bulldog	3	1.5	
	Shar Pei	1	0.5	
	Miniature Pinscher	1	0.5	
	St. Bernard	1	0,5	
	Spanish Mastiff	2	1	
Group 3: Terriers	Yorkshire Terrier	15	7.5	15.5
	Bodeguero Andaluz *	8	4	
	American Staffordshire Terrier	4	2	
	West Highland White Terrier	2	1	
	Staffordshire Bull Terrier	1	0.5	
	Bull Terrier	1	0.5	
Group 4: Dachshunds	Dachshund	3	1.5	1.5
Group 5: Spitz and primitive types	Siberian Husky	2	1	1
Group 6: Scent hounds and related breeds	Beagle	2	1	1
Group 7: Pointing dogs	Brittany Spaniel	2	1	4.5
	Saint Germain Pointer	1	0.5	
	English Setter	6	3	
Group 8: Retrievers, Flushing Dogs, Water Dogs	Labrador Retriever	6	3	9
	Golden Retriever	4	2	
	Spanish Water dog	2	1	
	English Cocker Spaniel	6	3	
Group 9: Companion and toy dogs	Shih Tzu	2	1	8.5
	Maltese	8	4	
	Lhasa Apso	3	1.5	
	Poodle	2	1	
	Pug	1	0.5	
	Bichon Frise	1	0.5	
Group 10: Sighthounds	Spanish Greyhound	3	1.5	1.5
Mixed-breed dogs		67	33.5	33.5

* Breed recognized by the Canine Real Society of Spain, but not by the FCI.

**Table 2 animals-11-03198-t002:** Descriptive analysis of the studied dog population.

Variable	C-BARQ (*n* = 377)		SAB (*n* = 32)	
	GP	CP	GP	CP
Sex				
Male	115 (52.5%)	79 (44.6%)	5 (38.5%)	5 (26.3%)
Female	85 (47.5%)	98 (55.4%)	8 (61.5%)	14 (73.7%)
Mean age (months)	65.6 (S.D. = 49.2)	38.8 (S.D. = 32.0)	57.2 (S.D. = 28.0)	38.2 (S.D. = 35.3)
Mean age at acquisition (months)	5.98 (S.D. = 14.1)	5.56 (S.D. = 13.6)	13.1 (S.D. = 19.5)	7.07 (S.D. = 20.4)
Neutered				
Yes	63 (31.5%)	24 (13.6%)	9 (69.2%)	5 (26.3%)
No	137 (68.5%)	153 (86.4%)	4 (30.8%)	14 (73.7%)
Activity pattern				
No activity	189 (94.5%)	140 (79.1%)	12 (92.3%)	15 (78.9%)
Agility/bike/walking/herding	11 (5.5%)	37 (20.9%)	1 (7.7%)	4 (21.1%)
Owner had more dogs				
No	69 (34.5%)	21 (11.9%)	7 (53.8%)	4 (21%)
One or more	131 (65.5%)	156 (88.1%)	6 (46.15%)	15 (78.9%)

Abbreviations: GP = general population; CP = Can de Palleiro breed; C-BARQ = Canine Behavioural Assessment and Research Questionnaire; SAB = Socially Acceptable Behaviour test.

**Table 3 animals-11-03198-t003:** Mean (with 95% confidence interval) scores obtained for general population (GP) dogs and Can de Palleiro breed (CP) for the different traits considered in the Canine Behavioural Assessment and Research Questionnaire.

C-BARQ Traits	GP			CP		
	Mean (Median, Q25–Q75)	95% CI		Mean (Median, Q25–Q75)	95% CI	
Stranger-directed aggression *	0.54 (0.40, 0–0.80)	0.44	0.63	0.72 (0.70, 0.20–1.10)	0.59	0.85
Owner-directed aggression *	0.30 (0.15, 0.20–0.37)	0.23	0.38	0.06 (0, 0–0)	0.01	0.12
Dog-directed aggression *	1.03 (0.75, 0.25–1.50)	0.89	1.17	1.30 (1.00, 0.50–1.75)	1.11	1.49
Dog-directed fear *	0.95 (0.75, 0.25–1.50)	0.83	1.07	0.51 (0.15, 0–1.25)	0.39	0.64
Familiar dog aggression	0.45 (0, 0–0.50)	0.34	0.56	0.53 (0.25, 0–1.00)	0.39	0.67
Trainability *	2.59 (2.60, 2.25–3.00)	2.51	2.67	2.87 (2.90, 2.50–3.40)	2.76	2.97
Chasing *	1.40 (1.25, 0.50–2.00)	1.23	1.56	2.32 (2.12, 1.62–3.12)	2.11	2.52
Stranger-directed fear *	0.56 (0.2, 0.32–1.00)	0.44	0.68	0.33 (0, 0–0.75)	0.23	0.44
Nons-ocial fear *	1.13 (1.00, 0.50,1.70)	1.02	1.25	0.70 (0.50, 0.17–1.00)	0.59	0.82
Separation-related problems *	0.80 (0.62, 0.25–1.37)	0.71	0.89	0.49 (0.38, 0–0.86)	0.40	0.59
Attachment/attention seeking	2.16 (2.06, 1.67–2.67)	2.05	2.27	2.03 (1.93, 1.50–2.67)	1.86	2.20
Excitability *	1.91 (2.00, 1.17–2.67)	1.79	2.03	1.45 (1.50, 1.00–2.00)	1.31	1.59
Energy *	2.25 (2.15, 1.50–3.00)	2.08	2.41	2.52 (2.50, 2.00, 3.00)	2.33	2.70
Touch sensitivity *	0.81 (0.50, 0–1.25)	0.68	0.94	0.58 (0.29, 0–0.81)	0.45	0.71

Abbreviations: C-BARQ = Canine Behavioural Assessment and Research Questionnaire; GP = general population; CP = Can de palleiro breed. * *p* < 0.05 when comparing GP and CP groups.

**Table 4 animals-11-03198-t004:** Results of an ordinal regression model for the effect of the breed (Can de Palleiro [CP] breed versus the general population [GP] of dogs) on behavioural traits, as measured by the Canine Behavioural Assessment and Research Questionnaire.

Variables	Owner-Directed Aggression		Dog-Directed Fear		Trainability		Chasing		Non-Social Fear		Separation-Related Problems		Excitability	
	Coefficient (95% CI)	*p*-Value	Coefficient (95% CI)	*p*-Value	Coefficient (95% CI)	*p*-Value	Coefficient (95% CI)	*p*-Value	Coefficient (95% CI)	*p*-Value	Coefficient (95% CI)	*p*-Value	Coefficient (95% CI)	*p*-Value
Breed CP *	0.18 (0.04–0.87)	0.033	0.43 (0.27–0.70)	0.001	2.56 (1.54–4.24)	<0.001	3.81 (2.45–5.93)	<0.001	0.42 (0.26–0.66)	<0.001	0.35 (0.20–0.59)	<0.001	0.48 (0.30–0.76)	0.002
Gender Male **	-	-	-	-	-	-	-	-	-	-	-	-	1.28 (0.84–1.96)	0.246
Neutered Yes ***	-	-	0.63 (0.37–1.97)	0.091	1.69 (1.01–2.84)	0.043	0.69 (0.43–1.11)	0.130	-	-	0.66 (0.37–1.15)	0.145	-	-
Age	0.99 (0.98–1.01)	0.095	-	-	-	-	1.00 (0.99–1.01)	0.360	0.99 (0.99–0.99)	0.014	-	-	-	-
Age at acquisition	-	-	-	-	-	-	-	-	-	-	-	-	0.98 (0.97–1.00)	0.063
First dog Yes ****	0.45 (0.12–1.66)	0.234	-	-	2.59 (1.54–4.44)	0.001	-	-	-	-	-	-	-	-
Activity Agility/bike/walking/ Herding *****	-	-	-	-	3.80 (1.74–8.29)	0.001	-	-	-	-	-	-	0.57 (0.28–1.17)	0.129

* GP was the base, ** Female was the base, ***/**** No was the base, ***** No activity was the base.

## Data Availability

The data presented in this study are available on request from the corresponding author.

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
