# Peer review of "Study of Behavioural Traits in Can de Palleiro (Galician Shepherd Dog)"

_animals, 2021, doi:10.3390/ani11113198_

Round 1

Reviewer 1 Report

The authors presented an interesting paper. One of the strongest points of this paper is that it works with a local dog breed, which is very rare to find among literature (even if some really ancient references may depict the earliest attempts to do it), even more to perform a behavioural study which will require fromt he authors for them to have a high ability to perform field tests, to objectively collect information but also to post process such information appropriately. Validation statsitical tools for observations of the scale must be used to make sure the behavioural constructs obtain are solidly and robustly quantified by the scale used and by the observers collecting data. Interobserver reliability testing may be of help.

Please found specific comments below.

Please be specific in the title. Indeed, authors are quite specific in the simple summary already when they address the “individual consistency in the capacity for behavioural response to stimuli and situations)”. This is what in behavioral science is known as copying strategies or styles. Indeed, copying is a very interesting trait to be selected in dogs with a functional attitude; shepherding, guiding, among others. However, after this, in the last sentence of the simple summary, authors address other rather complex traits (in terms that they may involve many cognitive processes) such as owner-direct aggression, dog-direct fear, excitability, non-social fear, and separation-related problems, trainability. I know the area is complex to report but an effort should be made in calling things by their name.

Abstract was very nice and I must say that working behavior from a selective point of view in endangered breeds is definitely not the easiest task to develop. I missed some numerical results and significance. All in all, what the reader reads at this point may determine whether he/she continues reading or not. Hence, an effort should be made to add more details (results) and the conclusions derived.

Keywords: Please try to avoid using the same words that are already present in the title. Search engines already search for title words, hence if you repeat, you lose a chance to be found and consequently cited.

Line 44. Start line without using The.

Try to avoid paragraphs of less than three sentences long.

Line 44 to 60. All this information and only one reference? Please support your information of previous bibliography. Even if these are ancient this are appropriate because provide sources for readers who may be interested in learning more about this endangered breed.

Objectives are expressed very generally while the outcomes of the paper are quite specific. Please be specific, refer to comments for title above.

Line 133 change resolve to solve.

I understand that Table may be extracted and modified from an external source, but it adds many dual concepts in the same definition which may not exactly refer to the same concept. Other things may need clarification, but this is one of the issues with behavioural science. Much needs to be explained. For instance,

How little acquainted a dog must be of a person for this person to be consider a stranger?

Threat and aggression is not the same, or were these used in the same manner in the Table.

What is a simple command? What would be a complex command?

By agitation we understand those unintentional and purposeless behaviors that result from feelings of inner restlessness. Applied to the specific context in which the term was used, it is the attention to others what startles the agitation response. How can we know that this indeed is not excitation? In my opinion and as it can be found in literature excitement is the state of being emotionally aroused while agitation is the act of agitating, or the state of being agitated; the state of being moved with violence, or with irregular action; commotion. If agitation involves a violent response, how can it be differenced from aggressivity? Hence, could you please define what you understand by agitation and add this a s a footnote. This apply to many concepts in this table which must be described and supported with references.

For instance, authors do provide this kind of explanations for instance when they define what is a neutral position for them in lines 152 to 154. I would like to see this all over the material and methods section, because this way we totally understand what the authors are measuring.

If any scales were used, in this case, it is not enough with referencing the scale. A little more must be said in order for the paper to be self-sufficient.

Table 3. What is a normal way?

What is unfamiliar and what familiar.

Section 2.3.

Was the proportional odds assumption tested? The key assumption in ordinal regression is that the effects of any explanatory variables are consistent or proportional across the different thresholds, hence this is usually termed the assumption of proportional odds (SPSS calls this the assumption of parallel lines but it’s the same thing). This is called the proportional odds assumption or the parallel regression assumption. Because the relationship between all pairs of groups is the same, there is only one set of coefficients (only one model). If this was not the case, we would need different models to describe the relationship between each pair of outcome groups. We need to test the proportional odds assumption, and we can use the tparallel option on the print subcommand. The null hypothesis of this chi-square test is that there is no difference in the coefficients between models, so we hope to get a non-significant result. The forementioned test indicates that we have not violated the proportional odds assumption. If the proportional odds assumption was violated, we may want to go with multinomial logistic regression. Considering scales were based on descriptions, how do we know proportion is kept from 1 to 2 and from 2 to 3 and so on.

Another concern for me is that although sample may be representative in the context of an endangered breed such as Can de Palleiro, quantitatively is samll. Considering the categorical nature of the traits, after all what we do here is to rank descriptions, normality may have very likely been violated. In this case mean does not really tell us anything. Median and quartiles would be more appropriate. Please check.

As this information is not clear I am sorry that I cannot progress further in the revision, otherwise it would not be fair. Hence, I would be willing to revise the paper once authors have revised it and then I can come to a decision basing on rather solid evidence.

Author Response

Comments and Suggestions for Authors

The authors presented an interesting paper. One of the strongest points of this paper is that it works with a local dog breed, which is very rare to find among literature (even if some really ancient references may depict the earliest attempts to do it), even more to perform a behavioural study which will require from the authors for them to have a high ability to perform field tests, to objectively collect information but also to post process such information appropriately. Validation statsitical tools for observations of the scale must be used to make sure the behavioural constructs obtain are solidly and robustly quantified by the scale used and by the observers collecting data. Interobserver reliability testing may be of help.

AU: Interobserver testing was performed, as indicated, and described in the appropriate sections (lines 157- 159 in Material and Methods. and 209-211 in Results).

Please found specific comments below.

Please be specific in the title. Indeed, authors are quite specific in the simple summary already when they address the “individual consistency in the capacity for behavioural response to stimuli and situations)”. This is what in behavioral science is known as copying strategies or styles. Indeed, copying is a very interesting trait to be selected in dogs with a functional attitude; shepherding, guiding, among others. However, after this, in the last sentence of the simple summary, authors address other rather complex traits (in terms that they may involve many cognitive processes) such as owner-direct aggression, dog-direct fear, excitability, non-social fear, and separation-related problems, trainability. I know the area is complex to report but an effort should be made in calling things by their name.

AU: Thanks for the insight; the authors consider that the best option to solve the mistake is changing the word temperament and the definition with the term “behavioural traits”.

Abstract was very nice and I must say that working behavior from a selective point of view in endangered breeds is definitely not the easiest task to develop. I missed some

numerical results and significance. All in all, what the reader reads at this point may determine whether he/she continues reading or not. Hence, an effort should be made to add more details (results) and the conclusions derived.

AU: More numerical results were included, although the changes introduced are partial due to the extension limitations of the journal itself.

Keywords: Please try to avoid using the same words that are already present in the title. Search engines already search for title words, hence if you repeat, you lose a chance to be found and consequently cited.

AU: Corrected according to the suggestion.

Line 44. Start line without using The.

AU: “The” was removed from the text.

Try to avoid paragraphs of less than three sentences long.

AU: The first sentence was joined to the next sentences.

Line 44 to 60. All this information and only one reference? Please support your information of previous bibliography. Even if these are ancient this are appropriate because provide sources for readers who may be interested in learning more about this endangered breed

AU: There is very little published about this breed and nothing in international magazines. We have added some more references, but their availability is limited. https://web.archive.org/web/20160303183101/http://mediorural.xunta.es/areas/gandaria/razas_autoctonas/canina/can_de_palleiro/

Objectives are expressed very generally while the outcomes of the paper are quite specific. Please be specific, refer to comments for title above.

AU: The objectives are now more specified, trying to standardize the terminology, as also indicated in the initial comments.

Line 133 change resolve to solve.

AU: Resolve was replaced with solve (line 135).

I understand that Table may be extracted and modified from an external source, but it adds many dual concepts in the same definition which may not exactly refer to the same concept. Other things may need clarification, but this is one of the issues with behavioural science. Much needs to be explained. For instance,

How little acquainted a dog must be of a person for this person to be consider a stranger?

Threat and aggression is not the same, or were these used in the same manner in the Table.

What is a simple command? What would be a complex command?

By agitation we understand those unintentional and purposeless behaviors that result from feelings of inner restlessness. Applied to the specific context in which the term was used, it is the attention to others what startles the agitation response. How can we know that this indeed is not excitation? In my opinion and as it can be found in literature excitement is the state of being emotionally aroused while agitation is the act of agitating, or the state of being agitated; the state of being moved with violence, or with irregular action; commotion. If agitation involves a violent response, how can it be differenced from aggressivity? Hence, could you please define what you understand by agitation and add this a s a footnote. This apply to many concepts in this table which must be described and supported with references.

For instance, authors do provide this kind of explanations for instance when they define what is a neutral position for them in lines 152 to 154. I would like to see this all over the material and methods section, because this way we totally understand what the authors are measuring.

If any scales were used, in this case, it is not enough with referencing the scale. A little more must be said in order for the paper to be self-sufficient.

AU: The authors considered to use C-BARQ questionnaire because it is a validated questionnaire for asses behavior traits in dogs. Indeed, some research demonstrated the validation of this tool (references were added in the text).

About stranger direct aggression, there are no more explanation about how acquainted a dog must be of a person for this person to be consider a strange in C-BARQ questionnaire or in the researches that used the C-BARQ, nevertheless, it seems that owner understand what stranger-direct aggression is because it seems that the stranger-directed subscale of the C-BARQ is a reliable tool to assess stranger-directed aggression in dogs (van der Berg et al., 2010).

The simple commands are cited in one by one in the C-BARQ questionnaire:

When off the leash, returns immediately when called, Obeys the “sit” command immediately, Obeys the “stay” command immediately… The items are scored as never, seldom, some-times, usually, always. The questionnaire doesn´t cited complex commands but make some questions about how easy it the dog to distract or to how easy he could learn some tricks (Serpell and Hsu, 2005).

The table 2 is an extract of the C-BARQ scoring that could be obtained in the C-BARQ web. Curiously, the questionnaire definition is more detailed and the word “agitation” is no present: “Some dogs show relatively little reaction to sudden or potentially exciting events and disturbances in their environment, while others become highly excited at the slightest novelty. Signs of mild to moderate excitability include increased alertness, movement toward the source of novelty, and brief episodes of barking. Extreme excitability is characterized by a general tendency to over-react. The excitable dog barks or yelps hysterically at the slightest disturbance, rushes toward and around any source of excitement, and is difficult to calm down.” So, we decided change agitation with excitation.

C-BARQ is a very long questionnaire that could be consulted in this web: https://vetapps.vet.upenn.edu/cbarq/. Authors considers that add C-BARQ to the present paper could make it too long. Nevertheless, factor and item structure of the C-BARQ was added as supplemental material (Duffy and Serpell, 2012).

· Serpell, J.A., Hsu, Y., 2005. Effects of breed, sex, and neuter status on trainability in dogs. Anthrozoös 18, 196–207.

· van den Berg, S. M., Heuven, H. C., van den Berg, L., Duffy, D. L., & Serpell, J. A. (2010). Evaluation of the C-BARQ as a measure of stranger-directed aggression in three common dog breeds. Applied animal behaviour science, 124(3-4), 136-141.

· Duffy, D. L., Serpell, J. A. Predictive validity of a method for evaluating temperament in young guide and service dogs. Applied Animal Behaviour Science. 2012, 138(1-2), 99-109.

Table 3. What is a normal way?

AU: Van der Borg et al, used the concept “normal speed” for “normal way”. The authors consider “normal way” as a neutral a approach to the dog in a normal speed, not too fast, not too slow.

What is unfamiliar and what familiar.

The researchs where SAB is used, didn´t define “unfamiliar” (Planta and Meester, 2007; De Meester et al., 2011; Van der Borg et al., 2010; van der Borg et al., 2017; Villa et al., 2017). Authors consider that a person is unfamiliar when the dog never has seen him or her before the test. Nevertheless, as the referee could see, the explanation that the authors gave of test is very similar that the explanation that was previous published in four of the research cited:

· Planta, J.U.D., De Meester, R. Validity of the Socially Acceptable Behavior (SAB) test as a measure of aggression in dogs to-wards non-familiar humans. Vlaams Diergeneeskundig Tijdschrift, 2007, 76(5), 359-368.

· De Meester, RH., De Bacquer, D., Peremans, K., Vermeire, S., Planta, D.J., Coopman, F., Audenaert, K. A preliminary study on the use of the Socially Acceptable Behavior test as a test for shyness/confidence in the temperament of dogs. J. Vet. Behav. Clin. Appl. Res. 2008, 3(4):161-170. doi:10.1016/j.jveb.2007.10.005.

· Van der Borg, J.A.M., Beerda, B., Ooms, M., de Souza, A.S., van Hagen, M., Kemp, B. Evaluation of behaviour testing for human directed aggression in

dogs. Appl. Anim. Behav. Sci. 2010, 128 (1-4), 78-90. doi:10.1016/j.applanim.2010.09.016

· Villa, PD., Barnard, S., Di Nardo, A., Iannetti, L., Vulpiani, M.P., Trentini, R., Serpell, J.A., Siracusa, C. Validazione del Socially Acceptable Behaviour (SAB) test su una popolazione di cani di proprietà del centro Italia. Vet Ital. 2017, 53 (1), 61-70. doi:10.12834/VetIt.321.1283.3 1.

· Van der Borg, J.A.M., Graat, E.A.M., Beerda, B. Behavioural testing based breeding policy reduces the prevalence of fear and aggression related behaviour in Rottweilers. Appl. Anim. Behav. Sci. 2017, 195 (2016), 80-86. doi:10.1016/j.applanim.2017.06.004

· De Meester, RH., Pluijmakers, J., Vermeire, S., Laevens, H. The use of the socially acceptable behavior test in the study of temperament of dogs. J. Vet. Behav. Clin. Appl. Res. 2011; 6(4), 211-224. doi:10.1016/j.jveb.2011.01.003

Section 2.3.

Was the proportional odds assumption tested? The key assumption in ordinal regression is that the effects of any explanatory variables are consistent or proportional across the different thresholds, hence this is usually termed the assumption of proportional odds (SPSS calls this the assumption of parallel lines but it’s the same thing). This is called the proportional odds assumption or the parallel regression assumption. Because the relationship between all pairs of groups is the same, there is only one set of coefficients (only one model). If this was not the case, we would need different models to describe the relationship between each pair of outcome groups. We need to test the proportional odds assumption, and we can use the tparallel option on the print subcommand. The null hypothesis of this chi-square test is that there is no difference in the coefficients between models, so we hope to get a non-significant result. The forementioned test indicates that we have not violated the proportional odds assumption. If the proportional odds assumption was violated, we may want to go with multinomial logistic regression. Considering scales were based on descriptions, how do we know proportion is kept from 1 to 2 and from 2 to 3 and so on.

AU: the propotional odd assumption was testes as we mentioned in the original version (Lines. 211-215, lines 186-189 in the new version). Yes, it is true that we did not make any mention in the results section, which is now included in the lines 248-249.

Another concern for me is that although sample may be representative in the context of an endangered breed such as Can de Palleiro, quantitatively is samll. Considering the categorical nature of the traits, after all what we do here is to rank descriptions, normality may have very likely been violated. In this case mean does not really tell us anything. Median and quartiles would be more appropriate. Please check.

We assume, as indicated, that the assumptions of normality will not be fulfilled, due to the type of distributions, to which the smallest sample in the SAB test is added. Therefore, we use non-parametric tests in the univariate comparison. Also in the descriptive table, we have added median and quartiles to complement the information.

Reviewer 2 Report

Review of: Study of Behavioural Characteristics of Can de Palleiro 2 (Galizian Shepherd dog)

I sincerely enjoyed this topic and I appreciate the need and wish to study this breed for many reasons. I am excited the authors are doing this.

However – I am concerned about a few points. Primarily that the comparison breed data is missing. The readers should know what dogs were used for comparison and I would argue you could run stats on breed groups as well.

The SAB data is likely too minimal (n) for stats to be meaningful and conclusions should be very limited with it.

Good luck.

Authors

There is no affiliation #5 yet there is a 6?

Summary and abstract – I would not say “to the best of our knowledge” in these parts – if you are unsure I would discuss that in the discussion not introduce in these two parts.

Is it truly reasonable to make conclusions from 19 CPs and 13 GPs – as the GP should at least equal the CP and – you are comparing CP to potentially 13 very different breeds and the abstract doesn’t say what they are – it could be 13 Chihuahuas … some detail is needed and I do worry that these numbers are very low compared to your C-BARQ data.

Intro

Line 44 – capitalize “can” … same comment in Table #6 & 7 and a few other places

Lines 44 – 56 – much of this needs to have references

Line 52 – what is the number ? 60 62 doesn’t make sense here

Lines 84 – 85 – consider not using observed and observer in the same sentence

Lines 90 – 92 are more discussion than intro

Lines 98 – 99 seems repetitive especially with questionnaire mentioned twice

Lines 99 – 100 – a questionnaire doesn’t contain “items”

Line 103 – ref the 1st in the list BEFORE the 2nd (17 before 20)

Materials and Methods

Line 112 – I do not know what “in its original initialism” means

Lines 118 – 119 – I hope you present these breeds in the results … must mentioning here in case you do not.

Table 1:

The SAB CP dogs are heavily female skewed and overall the SAB are skewed as not neutered – I assume this is discussed in the discussion

Check the table as you sometimes use “.” and you sometimes use “,” --- also – be consistent with this in all other tables as well bc they all differ a little

The ownership part could be explained better – I assume the “one or more” means more than 1 owner not more than 1 dog?

Table 2:

Not sure this is really needed – people could access the reference if needed? I feel similar about table 3 although that one seems more potentially needed.                                              

Line 147 – the term “accurately” is awkward here as why would anyone assume different?

Line 149 – analyze instead of analyse?

Table 3: (as tables should be able to stand alone shouldn’t you have a reference at the bottom of this? --- same for Tables 4 and 5 --- just a lot of info being published that is not yours but your references)

Do you mean sledge or sled? (#12)

Line 179 – what is “exceptional”

Results

Table 6

Personally – I would format Table 6 so that CP and GP were side by side instead of on top of each other – the way currently presented without lines makes it somewhat hard to follow

Line 214 – What breeds were these 2 dogs?

Line 217 – statistically what?

Lines 232 – 233 could be better explained (also – no period needed before the (Table 7)

Table 7

Has some formatting issues (titles stretch to 2 lines, etc.)

Discussion

The section is very short and in my opinion does not discuss many factors: what were the other breeds for example – that is very important piece of this – do they truly represent all dogs? Without info we cannot really compare or assume –

The gender differences and neuter differences should be better discussed as well.

References

Are not formatted in a consistent style – while Animals accepts most any style they ask for consistency and that was not completed and should have been.

Author Response

eview of: Study of Behavioural Characteristics of Can de Palleiro 2 (Galizian Shepherd dog)

I sincerely enjoyed this topic and I appreciate the need and wish to study this breed for many reasons. I am excited the authors are doing this.

However – I am concerned about a few points. Primarily that the comparison breed data is missing. The readers should know what dogs were used for comparison and I would argue you could run stats on breed groups as well.

AU: Information about breeds that were included in the study is now included in table 1 and in the discussion (lines 263-265).

The SAB data is likely too minimal (n) for stats to be meaningful and conclusions should be very limited with it.

AU: Despite that the small sample for SAB test, this study obtained significant results. Nevertheless, more studies are necessary with a mayor number of dogs tested with SAB test and it was added in the discussion as a limitation of the present research.

Authors

There is no affiliation #5 yet there is a 6?

AU: Affiliation 5 was changed with 6

Summary and abstract – I would not say “to the best of our knowledge” in these parts – if you are unsure I would discuss that in the discussion not introduce in these two parts.

AU: Line 26 The referee are right, we are sure that there are not previous studies about this breed so “to the best of our knowledge” was replaced with “interestingly”

Is it truly reasonable to make conclusions from 19 CPs and 13 GPs – as the GP should at least equal the CP and – you are comparing CP to potentially 13 very different breeds and the abstract doesn’t say what they are – it could be 13 Chihuahuas … some detail is needed and I do worry that these numbers are very low compared to your C-BARQ data.

AU: Details about the breed are added in the text (8 mixed breeds, 1 miniature Pinscher, 1 Dachshund, 1 Poodle, 1 German Shepherd, 1 Podenco). Also, the low number of the dogs performed C-BARQ was added that a limitation of the study in the discussion section.

Intro

Line 44 – capitalize “can” … same comment in Table #6 & 7 and a few other places

AU: Can is capitalized in line 44, and in the tables 1, 6 and 7 (now tables 2, 3 and 4).

Line 52 – what is the number ? 60 62 doesn’t make sense here

AU: The breed standard indicated literally about 60-62 cm. So, a dash is added between 60 and 62.

Lines 84 – 85 – consider not using observed and observer in the same sentence

AU: Observed is removed.

Lines 98 – 99 seems repetitive especially with questionnaire mentioned twice

AU: Questionnaire is replaced with “they” and the sentence was reconstructed: They are often used as a means of validating behavioural tests and typically provide detailed information about a given dog’s tendency to display different behaviours be-cause owners an observe their animals in a variety of situations over an extended period.

Lines 99 – 100 – a questionnaire doesn’t contain “items”

AU: Items was replaced with questions.

Line 103 – ref the 1st in the list BEFORE the 2nd (17 before 20)

AU: The references were changed

Materials and Methods

Line 112 – I do not know what “in its original initialism” means

AU: The authors think that the sentence was confused, so it was removed

Lines 118 – 119 – I hope you present these breeds in the results … must mentioning here in case you do not.

AU: Breeds are now present in the table 1.

Table 1:

The SAB CP dogs are heavily female skewed and overall the SAB are skewed as not neutered – I assume this is discussed in the discussion.

AU: Some lines were added to the discussion

Check the table as you sometimes use “.” and you sometimes use “,” --- also – be consistent with this in all other tables as well bc they all differ a little.

AU: “,” were replaced with “.”

The ownership part could be explained better – I assume the “one or more” means more than 1 owner not more than 1 dog?

AU: “Dogs previously owned” was changed by “the owner had more dogs”.

Table 2:

Not sure this is really needed – people could access the reference if needed? I feel similar about table 3 although that one seems more potentially needed.

AU: These tables were added as Appendix.

Line 147 – the term “accurately” is awkward here as why would anyone assume different?

AU: “Accurately” was removed

Line 149 – analyze instead of analyse?

AU: “Analyze” was replaced with “analyze”.

Table 3: (as tables should be able to stand alone shouldn’t you have a reference at the bottom of this? --- same for Tables 4 and 5 --- just a lot of info being published that is not yours but your references)

AU: References was added to the tables (now appendix) and in the text.

Do you mean sledge or sled? (#12)

AU: Sledge is the word used in SAB references (Planta and Meester, 2007; De Meester et al., 2011; Villa et al., 2017)

Line 179 – what is “exceptional”

AU: “i.e. bike, walking, herding” is added to the sentence.

Results

Table 6

Personally – I would format Table 6 so that CP and GP were side by side instead of on top of each other – the way currently presented without lines makes it somewhat hard to follow

AU: corrected, now table 3

Line 214 – What breeds were these 2 dogs?

AU: They are Can de Palleiro. A line is added in the text. Nevertheless, the results of the subtest performed by these dogs were analyzed.

Line 217 – statistically what?

“significant” was added to the sentence.

Lines 232 – 233 could be better explained (also – no period needed before the (Table 7)

AU: corrected and we tried to explain better now (lines 230-233).

Table 7

Has some formatting issues (titles stretch to 2 lines, etc.)

AU: corrected (now table 4)

Discussion

The section is very short and in my opinion does not discuss many factors: what were the other breeds for example – that is very important piece of this – do they truly represent all dogs? Without info we cannot really compare or assume –

AU: Information about the breed was added and the discussion was extended

The gender differences and neuter differences should be better discussed as well.

AU: Some sentences was added in order to better discuss gender and neuter differences.

References

Are not formatted in a consistent style – while Animals accepts most any style they ask for consistency and that was not completed and should have been.

AU: corrected

Round 2

Reviewer 1 Report

No further comments

Reviewer 2 Report

I feel that the authors responded appropriately to my prior concerns.